# Normally-Off p-GaN Gated AlGaN/GaN MIS-HEMTs with ALD-Grown Al_2_O_3_/AlN Composite Gate Insulator

**DOI:** 10.3390/membranes11100727

**Published:** 2021-09-23

**Authors:** Hsien-Chin Chiu, Chia-Hao Liu, Chong-Rong Huang, Chi-Chuan Chiu, Hsiang-Chun Wang, Hsuan-Ling Kao, Shinn-Yn Lin, Feng-Tso Chien

**Affiliations:** 1Department of Electronics Engineering, Chang Gung University, Taoyuan 333, Taiwan; r3287133@gmail.com (C.-H.L.); gain525252@gmail.com (C.-R.H.); wade03030333@gmail.com (C.-C.C.); smallflgt@hotmail.com (H.-C.W.); snoopy@mail.cgu.edu.tw (H.-L.K.); 2Department of Radiation Oncology, Chang Gung Memorial Hospital, Chang Gung University, Taoyuan 333, Taiwan; rt3126@gmail.com; 3Department of Medical Imaging and Radiological Sciences, College of Medicine, Chang Gung University, Taoyuan 333, Taiwan; 4Department of Electronics Engineering, Feng Chia University, Taichung 407, Taiwan; ftchien@fcu.edu.tw

**Keywords:** p-GaN E-mode HEMT, normally-off, gate insulator, lifetime, reliability

## Abstract

A metal–insulator–semiconductor p-type GaN gate high-electron-mobility transistor (MIS-HEMT) with an Al_2_O_3_/AlN gate insulator layer deposited through atomic layer deposition was investigated. A favorable interface was observed between the selected insulator, atomic layer deposition–grown AlN, and GaN. A conventional p-type enhancement-mode GaN device without an Al_2_O_3_/AlN layer, known as a Schottky gate (SG) p-GaN HEMT, was also fabricated for comparison. Because of the presence of the Al_2_O_3_/AlN layer, the gate leakage and threshold voltage of the MIS-HEMT improved more than those of the SG-HEMT did. Additionally, a high turn-on voltage was obtained. The MIS-HEMT was shown to be reliable with a long lifetime. Hence, growing a high-quality Al_2_O_3_/AlN layer in an HEMT can help realize a high-performance enhancement-mode transistor with high stability, a large gate swing region, and high reliability.

## 1. Introduction

High-electron-mobility transistors (HEMTs) are crucial for high-frequency and high-power applications because of their outstanding thermal properties, high breakdown fields, and high mobility levels. For fail-safe reasons, the power integrated circuits usually require high-performance normally-off devices [1]. Several strategies, such as the use of fluorine ion treatment [2], gate recess [3], and a p-type GaN cap layer [4,5], have been employed in normally-off HEMT devices. Normally-off devices fabricated with p-GaN gate technology offer low on-state resistance and large positive threshold voltage. However, to uniformly etch away the p-GaN in the non-gated access region and to overcome the plasma-induced damage during the p-GaN removal are extremely challenging tasks. Therefore, several teams are investigating the p-GaN etching process, such as etching stop technique [6], oxidation technique [7], and H plasma technique [8]. A positive threshold voltage can be obtained in HEMTs with p-type GaN layers, without affecting channel mobility; therefore, strategies involving the use of a p-type GaN layer are widely considered as the most effective among the strategies mentioned above. However, uniformly etching p-GaN away from the ungated access region and mitigating plasma-induced damage [9] during p-GaN removal are challenging. Cl_2_-based ion etching is typically used to remove the p-GaN layer. If the etching is imprecise, the surface becomes rough with plasma bumps and in-surface defects, which may cause gate lag [10]. To control the etching depth, an AlN etch stop layer [6] is inserted between the p-GaN and barrier layers. This helps achieve highly selective etching to obtain superior etching uniformity, lower gate leakage, and lower dynamic on-resistance. Usually, the low hole concentration of p-GaN limits the threshold voltage; when this occurs, the resulting product is unsatisfactory practical applications. Moreover, the p–n junction gate turns on at strong forward bias, leading to a considerably high gate leakage current. In this study, to minimize the defects that may arise after p-GaN etching and increase the gate swing region, we deposited Al_2_O_3_ and AlN layers [11] on the surface of the device through atomic layer deposition (ALD); these layers also improved the device performance, increased the turn-on voltage, reduced the gate leakage current, and increased the device stability at high temperatures.

## 2. Device Structure

The HEMT structures used for proposed devices were grown on 6-inch Si (111) substrates through metal–organic chemical vapor deposition (MOCVD). A 300-nm-thick undoped GaN channel was deposited on top of a 4-μm-thick undoped GaN buffer transition layer. Subsequently, a 12-nm-thick Al_0.17_Ga_0.83_N layer, a 1-nm-thick AlN layer, and a 70-nm-thick p-type GaN top layer were deposited. The Al_2_O_3_/AlN interlayer was deposited through ALD. A schematic representation of the device is shown in Figure 1a.

The device was thermally annealed in an MOCVD chamber at 720 °C for 10 min in a N_2_ atmosphere. The activated Mg concentration was determined to be 1 × 10^18^ /cm^3^ using the Hall measurement. The device was fabricated using mesa isolation with inductively coupled plasma (ICP) in the first step. Second, a p-GaN layer was etched through ICP etching with Cl_2_/BCl_3_/SF_6_ as the etching gas. When the mixed gas reached the AlN layer, the SF_6_ plasma [12] reacted with the Al atoms and formed a thin AlF_3_ etch stop layer. Subsequently, the sample was soaked in a diluted HF/NH_4_F solution to remove AlF_3_. Then, a Ti/Al/Ni/Au (25/120/25/150 nm) ohmic metal stack was deposited through electron beam (e-gun) evaporation to serve as the source and drain; the metal stack was annealed at 875 °C for 30 s in a N_2_ atmosphere in an RTA system, and the contact resistance of the MIS-HEMT and SG-HEMT were 7.6 × 10^−6^ and 8.1 × 10^−6^ Ω·cm^2^, respectively. The oxide/insulator layer Al_2_O_3_/AlN (9/2.5 nm) was deposited through ALD, as shown in Figure 1b. In addition, Trimethylaluminum, O_2_, and N_2_ were used as metal precursors, O and N source, respectively. The RF power and chamber temperature were 60 W and 300 ℃. Then, an ohmic via was etched using buffered oxide etching. Ti/Au (25/120 nm) layers were then deposited through e-gun evaporation to serve as gate electrodes. By contrast, the gate metal of the SG-HEMT was just a deposited Ti/Au (25/120 nm) layer.

Figure 2 shows the conduction bands of the MIS-HEMT and SG-HEMT, which were obtained through TCAD simulation. The aim of this study was to increase V_TH_ and the gate voltage swing of the device by using Al_2_O_3_/AlN to increase the conduction band. The 2DEG height of the MIS-HEMT was more than that of the SG-HEMT, which resulted in the MIS-HEMT exhibiting better characteristics than the SG-HEMT.

## 3. Experimental Result and Discussion

For examining the effect of the Al_2_O_3_/AlN/p-GaN interface on device performance and for investigating the relationship of the AlN layer formed through ALD with an AlN etch stop layer at the sidewall and ungated region, we measured the I–V characteristics of the devices. Because an Al_2_O_3_/AlN layer can fill the nitrogen vacancies on the surfaces of p-GaN and AlN, depositing an Al_2_O_3_/AlN layer had the same effect as passivation. The I–V curve revealed that the gate leakage current was suppressed. Furthermore, because the polarization effect was enhanced due to the presence of the Al_2_O_3_/AlN layer, the MIS-HEMT had a higher drain current density and on/off ratio than the SG-HEMT. The off-state drain currents of the MIS-HEMT and SG-HEMT were 6 × 10^−6^ and 2 × 10^−4^ mA/mm, respectively, at V_GS_ = 0 V and V_DS_ = 10 V, as shown in Figure 3a. The I_DS_–V_DS_ characteristics are shown in Figure 3b. The saturated drain currents of the MIS-gate and metal-gate devices were 363 and 247 mA/mm at V_DS_ = 10 V. Figure 3c shows the MIS-gate HEMT data. The figure indicates that the Al_2_O_3_/AlN layer had a high gate voltage swing of more than 20 V [13]. To determine the device reliability, we measured off-state breakdown voltage; when measuring the off-state breakdown voltage, the device gate must be biased to 0 V to ensure that the device is in the off state. The results are plotted in Figure 3d. Because the Al_2_O_3_/AlN layer exhibited an effect similar to passivation and minimized any damage that might have been caused by etching, the Al_2_O_3_/AlN gate device had a higher off-state breakdown voltage of 656 V.

To analyze the higher off-state breakdown voltage of the MIS-HEMT, we simulated the electric field distributions of the MIS-HEMT and SG-HEMT using TCAD. Figure 4 shows the schematic cross-sections of the two devices. The electric field was always concentrated at the gate edge in the drain side, and the electric field distributions of both devices were simulated at V_GS_ = 0 V and V_DS_ = 650 V. A comparison of the two devices shows that the electric field of the MIS-HEMT was low at the gate edge owing to the electric field being dispersed by the insulator layer. Negative charges appeared at the interface between the insulator and AlGaN owing to the negative polarization, which reduced the electric field near the gate edge at the drain side [14]. Consequently, MIS-HEMT had a higher off-state breakdown voltage due to the presence of the AlN/Al_2_O_3_ stack.

Typical C-V characteristics were measured at 500 kHz, as shown in Figure 5. As the gate bias increases above a certain voltage, the capacitance value starts increasing and saturates. MIS-HEMT showed higher capacitance than SG-HEMT due to the Al_2_O_3_/AlN stacking. Figure 6a shows the dynamic on-resistance to static on-resistance ratio (R_Dynamic, on_, R_Static,on_) at different quiescent gate voltages. There are two bias conditions that must be considered: pulse voltage (V_GSP_, V_DSP_) and quiescent voltage (V_GSQ_, V_DSQ_). During the measurement, when the pulse voltage switched to the quiescent voltage rapidly with a 2 μs pulse width and 200 μs period, the charge carriers were trapped by defects. The quiescent gate bias was swept from 0 to −15 V in increments of −5 V. As seen in the figure, the MIS-HEMT had a lower dynamic Ron than the SG-HEMT; the MIS-HEMT had a dynamic Ron of 1.18, whereas the SG-HEMT had a dynamic Ron of 2.13. Therefore, the MIS-HEMT had a lower surface trap density surface, which inhibited current collapse. In this study, forward TDDB measurements were performed on the MIS-HEMT to assess the strength of the Al_2_O_3_/AlN gate dielectric. Figure 6b shows the variation in gate current during the TDDB experiments at three different gate voltages (14, 15, and 16 V). The gate current was initially low, owing to the accumulation of negative charges under the gate. Then, the gate leakage current became noisy owing to the formation of a percolation path. This was followed by a sudden increase in the gate leakage current owing to the hard breakdown of the gate dielectric [15]. A Weibull plot was drawn based on the time-to-breakdown distributions. The slope shows a tight distribution and less variability, as shown in Figure 6c. Moreover, lifetimes of 1% and 63.2% were investigated and are shown in Figure 6d. The gate leakage mechanisms at the measured gate bias region were determined to be 8.5 V at a failure rate of 63.2% and 6.1 V at a failure rate of 1%. Therefore, the MIS p-GaN can effectively suppress the trapping/detrapping behavior and then enhance the gate reliability. Figure 7 compares the V_G_BD_–I_G_ characteristics of the MIS-HEMT with that of other reported p-GaN gate HEMTs [5,16,17,18,19,20,21,22]. The MIS-HEMT fabricated in this study exhibited the highest V_G_BD_ and lowest gate leakage current.

## 4. Conclusions

A normally-off p-GaN HEMT with an Al_2_O_3_/AlN layer was investigated. The device exhibited excellent I–V characteristics. The turn-on voltage was higher than 20 V, with a threshold voltage of 3 V and a high saturation drain current of approximately 363 mA/mm. Moreover, the gate leakage current and instability were suppressed. The device off-state breakdown voltage in the MIS gate was increased because of the presence of the Al_2_O_3_/AlN layer deposited using ALD; this layer helped form an excellent interface between p-GaN and AlN. The lower trap density and trapping/detrapping effects were analyzed using pulse measurement. Overall, an Al_2_O_3_/AlN dielectric gate layer considerably improves the performance, reliability, and stability of HEMTs. Therefore, the as-developed design has a high potential for use in the manufacture of normally-off p-GaN gate HEMTs, which would allow them to be used in practical applications.

## Figures and Tables

**Figure 1 membranes-11-00727-f001:**
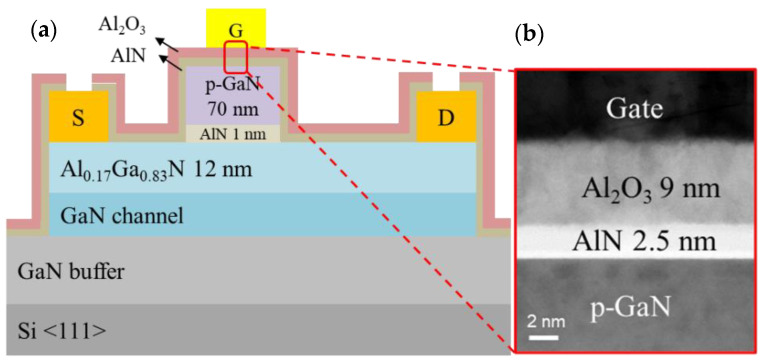
(**a**) Schematic of the device structure with L_GS_/L_G_/L_GD_/W_G_ = 2/4/10/100 µm. (**b**) transmission electron microscopy (TEM) photograph of the device.

**Figure 2 membranes-11-00727-f002:**
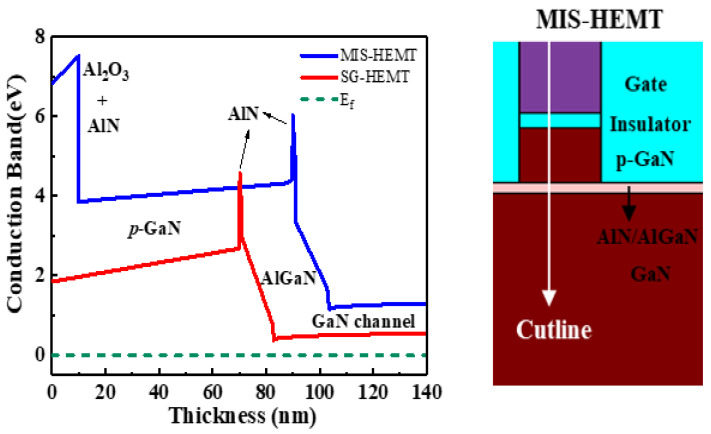
Energy band diagrams of schottky-gated high electron mobility transistor (SG-HEMT) and metal–insulator–semiconductor-high electron mobility transistor (MIS-HEMT) obtained through TCAD simulation.

**Figure 3 membranes-11-00727-f003:**
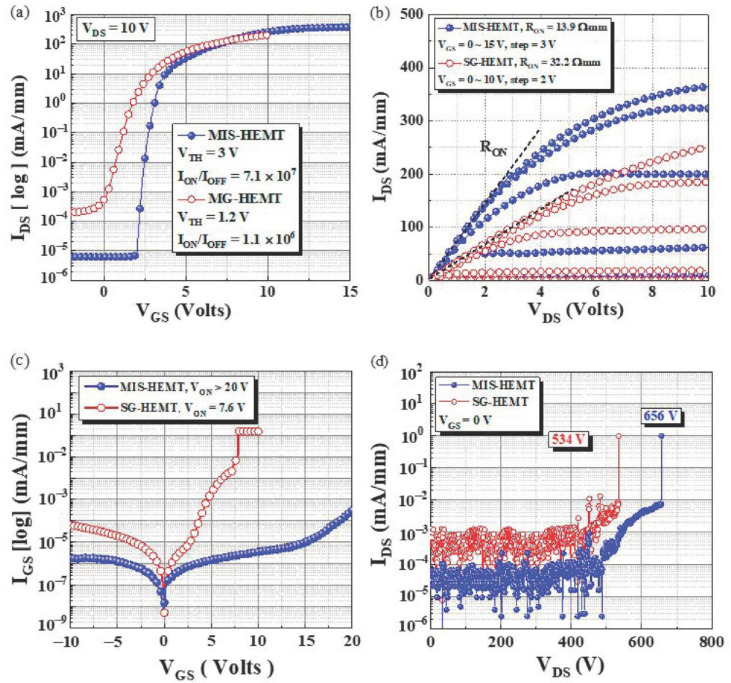
(**a**) Log-scale I_DS_–V_GS_ transfer characteristics, (**b**) I_DS_–V_DS_ output characteristics, (**c**) I_GS_–V_GS_ characteristics, and (**d**) off-state breakdown voltages of the MIS-HEMT and SG-HEMT.

**Figure 4 membranes-11-00727-f004:**
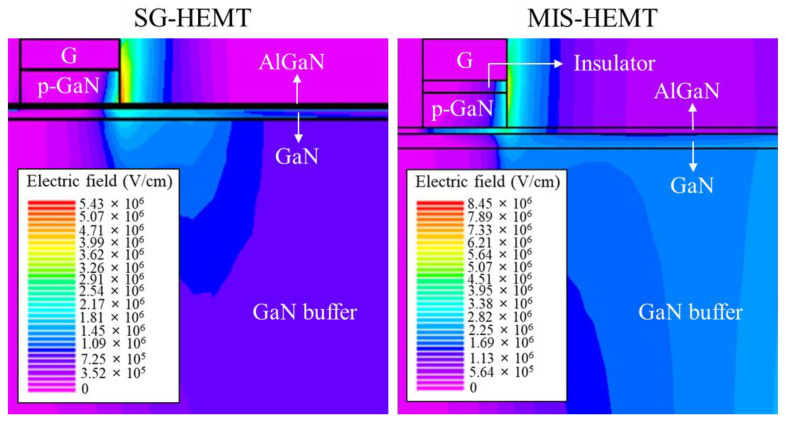
Electric field distributions of the SG-HEMT and MIS-HEMT simulated using TCAD.

**Figure 5 membranes-11-00727-f005:**
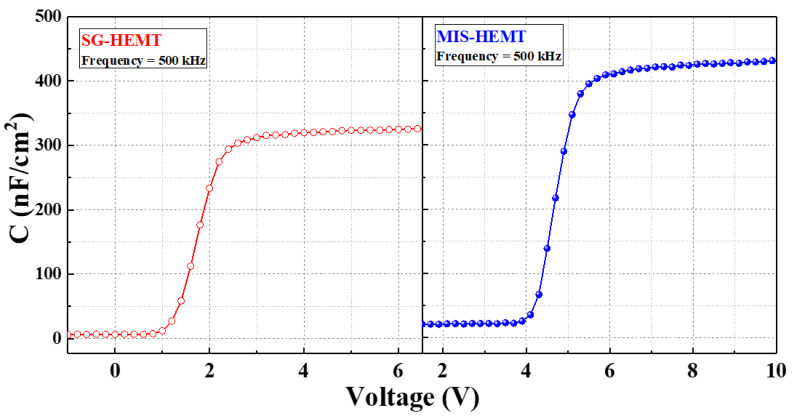
C–V characteristics of SG-HEMT and MIS-HEMT.

**Figure 6 membranes-11-00727-f006:**
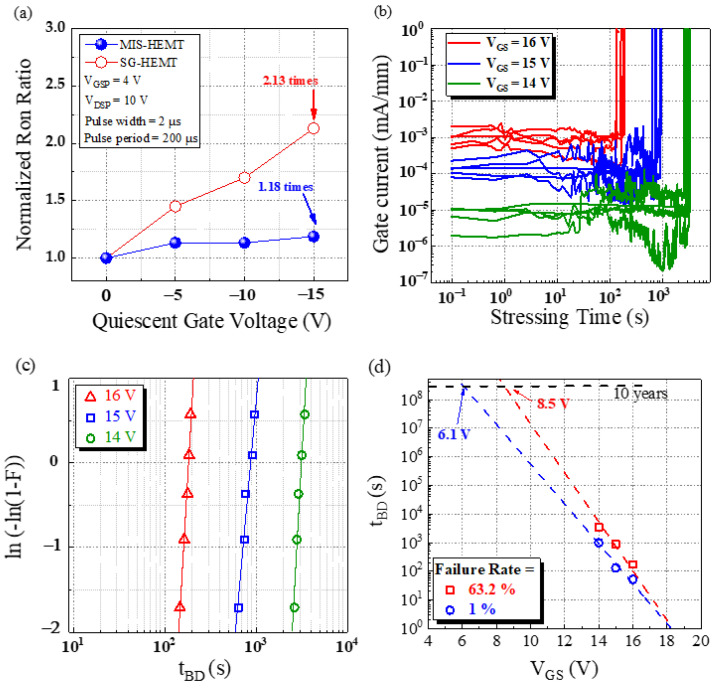
(**a**) Gate lag behavior and dynamic Ron, (**b**) TDDB gate current with three different gate voltages, (**c**) Weibull plot of the time-to-breakdown distributions, and (**d**) lifetime prediction with failure rate of 63.2% and 1% of MIS-HEMT.

**Figure 7 membranes-11-00727-f007:**
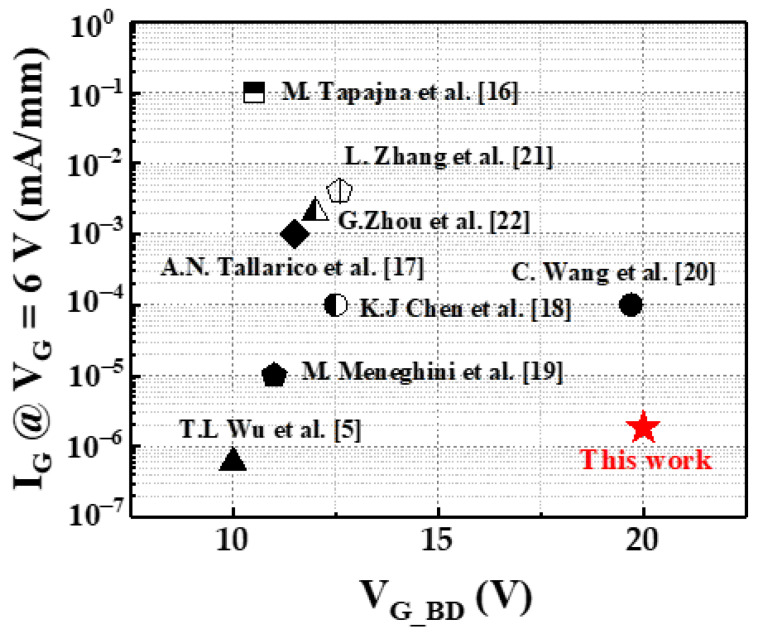
Comparison between the V_G_BD_–I_G_ characteristics of the proposed device and those of other devices [5,16,17,18,19,20,21,22].

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
