# Peer review of "Normally-Off p-GaN Gated AlGaN/GaN MIS-HEMTs with ALD-Grown Al2O3/AlN Composite Gate Insulator"

_membranes, 2021, doi:10.3390/membranes11100727_

Round 1

Reviewer 1 Report

The authors report on characterization of normally-off p-GaN/AlGaN/GaN HEMTs processed with Al2O3/AlN gate dielectric. Although the results are interesting, no details on the ALD process of the gate dielectric growth are given and the analysis of interface states density is not physically sound (see below my detail comments). Moreover, the manuscript is written in poor English making the statements difficult to follow. Therefore, the manuscript cannot be accepted in the current form.

  1. The authors are asked to give more details on ALD process of the Al2O3/AlN gate stack as well as possible surface treatment before the deposition. In fact, it is clear/convincing how they achieved the growth of the oxide on top of the nitride dielectric. Further, the chemical composition of the gate stack needs to be confirmed by e.g. XPS analysis.
  2. The part related to determination of interface state density Dit seems to be completely misleading. First, for SG-HEMT, it is not clear where the interface traps are located. Second, for evaluation of Dit using frequency dependent CV measurements, the applied gate voltage where the dispersion is measured must be set so that the occupation of interface states is modified during the measurements. This means, that the Fermi level at the interface (where Dit is measured) crosses the interface traps which are able to respond to AC signal, i.e. their emission time is similar or shorter than the time corresponding to 1/f. For the frequency and temperature used in the experiment, these traps can be located < 0.5 eV below the conduction band edge. Looking at the band diagram shown in Fig. 2, it seems that interface traps at dielectric/p-GaN interface are too slow (located deeper in the bandgap) to emit/capture electrons from the conduction band. Therefore, the given value of Dit does not provide the measure of the interface trap density.
    The authors are therefore advised to either discuss the Dit analysis in detail or remove this paragraph completly.
  3. The authors are asked to discuss the value of the capacitance at the plateau, i.e. at at VG > 2.5 and 6 V for SG- and MOS-HEMT, respectively. In fact, lower total capacitance in the plateau is expected for MOS-HEMT compared to SG-HEMT, in clear contrast to the observed CV curve.
  4. Following the previous comment, the band diagram in Fig. 2 suggest quite low hole concentration in the p-GaN (straight line of the bands). Is the p-GaN fully depleted? Does the (depletion) thickness of the p-GaN correspond to the measured total capacitance shown in Fig. 5. This needs to be discussed and clarified.
  5. In Fig. 2, the figures (a) and (b) are the same as those shown in Fig. 3(a) and (b). Please correct.
  6. Description of the pulsed measurements for dynamic Ron determination is hard to follow. For instance, I do not understand the sentence: “When the pulse voltage switched to the quiescent voltage rapidly, after measurement, the dynamic Ron to static Ron ratio of the MIS-HEMT and SG-HEMT were switched with a 2 μs pulse width and 200 μs period.” Please reword.
  7. Finally, the manuscript is written by poor English, so that some statements are unclear or even dubious. Just a few examples: “HEMT had a higher drain current density and on/off ratio than the SG-HEMT, that MIS-HEMT has approximately two orders more than SG-HEMT.”; “Figure 3(c) shows the MIS-gate HEMT data. The figure indicates that the Al2O3/AlN layer had a high gate voltage swing of more than 20 V” – what data are shown in Fig. 3(c)? More than 20 V in respect to what?

Author Response

Reviewer 1
1. The authors are asked to give more details on ALD process of the Al2O3/AlN gate stack as well as possible surface treatment before the deposition. In fact, it is clear/convincing how they achieved the growth of the oxide on top of the nitride dielectric. Further, the chemical composition of the gate stack needs to be confirmed by e.g. XPS analysis.

Ans: Thanks for your suggestion. In this work, the Al2O3/AlN was stacked after the clean process by chemical solution and the annealing process by RTA system. Due to limited time for XPS arrangement thus we didn’t have the XPS analysis, but the Al2O3/AlN stack at gate surface can be confirmed in the reference [1]. In this article, Al2O3 was stacked by ALD and there is XPS analyze to confirm the chemical composition. We adopted the same equipment to demonstrate the ALD process in our work. 

[1] C. -Y. Liu et al., "High-Performance Ultraviolet 385-nm GaN-Based LEDs With Embedded Nanoscale Air Voids Produced Through Atomic Layer Deposition and Al2O3 Passivation," in IEEE Electron Device Letters, vol. 37, no. 4, pp. 452-455, April 2016, doi: 10.1109/LED.2016.2532352.

  1. The part related to determination of interface state density Dit seems to be completely misleading. First, for SG-HEMT, it is not clear where the interface traps are located. Second, for evaluation of Dit using frequency dependent CV measurements, the applied gate voltage where the dispersion is measured must be set so that the occupation of interface states is modified during the measurements. This means, that the Fermi level at the interface (where Dit is measured) crosses the interface traps which are able to respond to AC signal, i.e. their emission time is similar or shorter than the time corresponding to 1/f. For the frequency and temperature used in the experiment, these traps can be located < 0.5 eV below the conduction band edge. Looking at the band diagram shown in Fig. 2, it seems that interface traps at dielectric/p-GaN interface are too slow (located deeper in the bandgap) to emit/capture electrons from the conduction band. Therefore, the given value of Dit does not provide the measure of the interface trap density.

Ans: Thanks for your suggestion and the C-V measurement and the Dit analyze have been removed.

  1. The authors are therefore advised to either discuss the Dit analysis in detail or remove this paragraph completly.
    The authors are asked to discuss the value of the capacitance at the plateau, i.e. at at VG > 2.5 and 6 V for SG- and MOS-HEMT, respectively. In fact, lower total capacitance in the plateau is expected for MOS-HEMT compared to SG-HEMT, in clear contrast to the observed CV curve.

Ans: Thanks for your suggestion and the C-V measurement and the Dit analyze have been removed.

4.Following the previous comment, the band diagram in Fig. 2 suggest quite low hole concentration in the p-GaN (straight line of the bands). Is the p-GaN fully depleted? Does the (depletion) thickness of the p-GaN correspond to the measured total capacitance shown in Fig. 5. This needs to be discussed and clarified.

Ans: Thanks for your careful reading. In Fig. 2, the band diagram can not observe the hole concentration and the straight line in the structure is meaning band diagram is calculated from this line. In addition, the conduction band of both device at channel region is above the fermi-level, so it fully depleted the 2DEG.

5.In Fig. 2, the figures (a) and (b) are the same as those shown in Fig. 3(a) and (b). Please correct.

Ans: Thanks for your careful reading. Fig. 2 shows the conduction bands of the MIS-HEMT and SG-HEMT, which were obtained through TCAD simulation and Fig. 3(a), (b) shows the DC characteristics of both devices.

6.Description of the pulsed measurements for dynamic Ron determination is hard to follow. For instance, I do not understand the sentence: “When the pulse voltage switched to the quiescent voltage rapidly, after measurement, the dynamic Ron to static Ron ratio of the MIS-HEMT and SG-HEMT were switched with a 2 μs pulse width and 200 μs period.” Please reword.

Ans: Thanks for your suggestion and the sentence has been revised.

7.Finally, the manuscript is written by poor English, so that some statements are unclear or even dubious. Just a few examples: “HEMT had a higher drain current density and on/off ratio than the SG-HEMT, that MIS-HEMT has approximately two orders more than SG-HEMT.”; “Figure 3(c) shows the MIS-gate HEMT data. The figure indicates that the Al2O3/AlN layer had a high gate voltage swing of more than 20 V” – what data are shown in Fig. 3(c)? More than 20 V in respect to what?

Ans: Thanks for your careful reading. The sentence “HEMT had a higher drain current density and on/off ratio than the SG-HEMT, that MIS-HEMT has approximately two orders more than SG-HEMT.” Has been revised. However, the IGS-VGS characteristic of both devices and the “gate voltage swing” means gate operation range of device. Therefore, the MIS-HEMT shows the low gate leakage current when the gate is operated at 20V, as shown in Fig 3(c).

Reviewer 2 Report

This manuscript investigated normally-off p-GaN MIS-HEMTs with ALD grown gate dielectrics. Although it is an interesting work, the current manuscript still needs improvements.

  1. The title is not appropriate since high gate voltage swing is just one property of this device. It should be revised, for example, “Normally-off p-GaN gated AlGaN/GaN MIS-HEMTs with ALD-grown Al2O3/AlN composite gate insulator”.
  2. Recently, there is a new plasma-based method to realize E-mode GaN HEMTs without etching. In order to provide a complete context, the authors should also consider adding some references on this method in the introduction part, such as R. Hao et al., Appl. Phys. Lett. 109, 152106 (2016); C. Yang et al., IEEE Electron Device Letters 42 (8), 1128-1131, 2021.
  3. How is Dit calculated? Some equations may be needed.
  4. Have the authors tried different thickness combinations for Al2O3/AlN layer? Please comment on whether different thicknesses have an impact on the device performance.
  5. The references used in the benchmark plot in Fig. 7 are not adequate. Please add more recent references for comparison.
  6. The resolution of Fig. 2-7 is low. Please replace them with high resolution figures.
  7. There are some grammatical errors and typos. For example, the first word in the introduction part should be capitalized. Please carefully proofread the manuscript.

Author Response

Reviewer2

1.The title is not appropriate since high gate voltage swing is just one property of this device. It should be revised, for example, “Normally-off p-GaN gated AlGaN/GaN MIS-HEMTs with ALD-grown Al2O3/AlN composite gate insulator”.

Ans: Thanks for your suggestion and the title has been revised.

2.Recently, there is a new plasma-based method to realize E-mode GaN HEMTs without etching. In order to provide a complete context, the authors should also consider adding some references on this method in the introduction part, such as R. Hao et al., Appl. Phys. Lett. 109, 152106 (2016); C. Yang et al., IEEE Electron Device Letters 42 (8), 1128-1131, 2021.

Ans: Thanks for your suggestion and the p-GaN etching process methods were added in the manuscript.

3.How is Dit calculated? Some equations may be needed.

Ans: Thanks for your suggestion. We follow suggestion of question 3 from Review1, so the C-V measurement and the Dit analyze have been removed.

4.Have the authors tried different thickness combinations for Al2O3/AlN layer? Please comment on whether different thicknesses have an impact on the device performance.

Ans: Thanks for your suggestion. We didn’t have the experiment of different thickness for Al2O3/AlN layer. The thicker AlN may cause the over lattice stress, so the AlN layer just approximately 2nm. In addition, the Al2O3 is relate to the VTH which increasing with the Al2O3 thickness.

5.The references used in the benchmark plot in Fig. 7 are not adequate. Please add more recent references for comparison.

Ans: Thanks for your suggestion and we have added more reference in Fig. 7 [24], [25].

  1. Zhang, Z. Zheng, S. Yang, W. Song, J. He and K. J. Chen, "p-GaN Gate HEMT With Surface Reinforcement for Enhanced Gate Reliability," in IEEE Electron Device Letters, vol. 42, no. 1, pp. 22-25, Jan. 2021, doi: 10.1109/LED.2020.3037186.
  2. Zhou et al., "Gate Leakage Suppression and Breakdown Voltage Enhancement in p-GaN HEMTs Using Metal/Graphene Gates," in IEEE Transactions on Electron Devices, vol. 67, no. 3, pp. 875-880, March 2020, doi: 10.1109/TED.2020.2968596.

6.The resolution of Fig. 2-7 is low. Please replace them with high resolution figures.

Ans: Thanks for your suggestion and we have optimized the quality of these graphs.

7.There are some grammatical errors and typos. For example, the first word in the introduction part should be capitalized. Please carefully proofread the manuscript.

Ans: Thanks for your suggestion and we have revised the errors in this manuscript.

Round 2

Reviewer 1 Report

The authors addressed most of my comments. However, no details on ALD growth of the Al2O3/AlN gate stack have been added (as proposed in original comment No. 1). Clearly, introduction of the MOS gate stack onto p-GaN/AlGaN/GaN HEMT represents the main novelty of the paper. Therefore, the reader should be informed in detail on the preparation of the gate stack. Therefore, I insist on my original comment and ask the authors to add more detail on ALD process of Al2O3/AlN gate stack (deposition temperature, sequence of deposition, precursors etc.).

Author Response

Ans: Thanks for your suggestion and the condition of ALD process has been added in the manuscript. The Al2O3/AlN layer was deposited by plasma-enhanced ALD. In addition, Trimethylaluminum, O2 and N2 was used as a metal precursor, O and N sources, respectively. The RF power and chamber temperature were 60 W and 300 ℃. More information in attached file.

Reviewer 2 Report

The authors didn't address most of the reviewers' concerns satisfactorily, and only minor changes are made in the revised manuscript. For the critical concern on C-V and Dit, instead of directly addressing it, the authors removed them completely, which is unacceptable. C-V and Dit analysis are essential for the new design of gate dielectrics.  

Author Response

Ans: Thanks for your suggestion. The C-V characteristics for both device were added in the manuscript and they were measured at 500kHz. However, to calculate the Dit precisely, it is necessary to measure C-V with different frequency at different temperatures, due to tight revision duration, we may not to make a new process lot to re-measure it. Therefore, the Dit is was removed and we will extract precise Dit in the near future. More information in attached file

Round 3

Reviewer 2 Report

The reviewer has no additional comments.